# Identification of the Large-Conductance Ca^2+^-Regulated Potassium Channel in Mitochondria of Human Bronchial Epithelial Cells

**DOI:** 10.3390/molecules26113233

**Published:** 2021-05-27

**Authors:** Aleksandra Sek, Rafal P. Kampa, Bogusz Kulawiak, Adam Szewczyk, Piotr Bednarczyk

**Affiliations:** 1Laboratory of Intracellular Ion Channels, Nencki Institute of Experimental Biology, Polish Academy of Sciences, 02-093 Warsaw, Poland; a.sek@nencki.edu.pl (A.S.); r.kampa@nencki.edu.pl (R.P.K.); b.kulawiak@nencki.edu.pl (B.K.); a.szewczyk@nencki.edu.pl (A.S.); 2Faculty of Chemistry, University of Warsaw, 02-093 Warsaw, Poland; 3Department of Physics and Biophysics, Institute of Biology, Warsaw University of Life Sciences—SGGW, 02-776 Warsaw, Poland

**Keywords:** mitochondria, mitoBK_Ca_ channel, human bronchial epithelial cells, potassium channel modulators, NS11021, paxilline

## Abstract

Mitochondria play a key role in energy metabolism within the cell. Potassium channels such as ATP-sensitive, voltage-gated or large-conductance Ca^2+^-regulated channels have been described in the inner mitochondrial membrane. Several hypotheses have been proposed to describe the important roles of mitochondrial potassium channels in cell survival and death pathways. In the current study, we identified two populations of mitochondrial large-conductance Ca^2+^-regulated potassium (mitoBK_Ca_) channels in human bronchial epithelial (HBE) cells. The biophysical properties of the channels were characterized using the patch-clamp technique. We observed the activity of the channel with a mean conductance close to 285 pS in symmetric 150/150 mM KCl solution. Channel activity was increased upon application of the potassium channel opener NS11021 in the micromolar concentration range. The channel activity was completely inhibited by 1 µM paxilline and 300 nM iberiotoxin, selective inhibitors of the BK_Ca_ channels. Based on calcium and iberiotoxin modulation, we suggest that the C-terminus of the protein is localized to the mitochondrial matrix. Additionally, using RT-PCR, we confirmed the presence of α pore-forming (Slo1) and auxiliary β3-β4 subunits of BK_Ca_ channel in HBE cells. Western blot analysis of cellular fractions confirmed the mitochondrial localization of α pore-forming and predominately β3 subunits. Additionally, the regulation of oxygen consumption and membrane potential of human bronchial epithelial mitochondria in the presence of the potassium channel opener NS11021 and inhibitor paxilline were also studied. In summary, for the first time, the electrophysiological and functional properties of the mitoBK_Ca_ channel in a bronchial epithelial cell line were described.

## 1. Introduction

Human bronchial epithelial (HBE) cells play a fundamental role as a defensive barrier supporting the maintenance of proper airway function. The monolayer of HBE cells creates a connection between the external and internal environment in the body, which makes them the main target of inhaled insults [1]. Epithelial cells are rich in a wide variety of ion transport proteins, among which Na^+^, Cl^−^ and K^+^ channels have been identified to play a role in the regulation of the fluid layer [2]. Malfunctions of bronchial epithelial ions transport processes and the subsequent impairment of the liquid balance in the lungs are associated with severe diseases, such as cystic fibrosis and pulmonary edema [3,4].

Mitochondrial channels maintain ion homeostasis in mitochondria, are responsible for changes in the volume of the interior of the mitochondria, and participate in the creation of a pH gradient [5]. Potassium channels (mitoK) similar to those in the plasma membrane are also present in the inner mitochondrial membrane. Potassium ion transport, known as the potassium cycle, is essential for the optimal course of oxidative phosphorylation and affects the mitochondrial membrane potential (∆Ψ). The influx of K^+^ into the mitochondria contributes to the change in matrix volume, regulates the β-oxidation of fatty acids and increases the rate of mitochondrial respiration [6,7]. In recent years, many hypotheses have been proposed that potassium channels present in the inner mitochondrial membrane induce the ability to reduce damage to cells exposed to a harmful factor [8,9]. Activation of potassium channels may lead to the induction of cytoprotection against ischemia/reperfusion damage in various cells [10,11].

Bronchial epithelial cells synthetize excess reactive oxygen species (ROS) as a result of an imbalance between oxidants and antioxidants, which contributes to chronic pneumonia and damage to lung tissue [1]. Clear evidence indicates that high doses of mitochondrial ROS may promote apoptosis; however, ROS are also involved in cell survival signaling [12,13]. It has been proposed that activation of mitoK by potassium channel openers mediating K^+^ influx into the mitochondrial matrix may reduce ROS synthesis [14].

Mitochondrial large-conductance Ca^2+^-regulated potassium (mitoBK_Ca_) channels were originally identified in the human glioma cell line LN229 using the patch-clamp technique [15]. Further studies also showed the presence of this channel in mitochondria isolated from the brain [16,17], skeletal muscle [18], heart [19,20], endothelial cells [21] and dermal fibroblasts [22]. Depending on the type of cell in which the BK_Ca_ channel exists, various splice variants have been observed [23]. Mitochondrial BK_Ca_ channels possess pharmacological properties similar to those described for plasma membrane BK_Ca_ channels. The channel is regulated by the membrane voltage and endogenous factors such as calcium, heme and hemin, and carbon monoxide [15,24,25,26]. This mitochondrial channel is activated by the potassium channel openers NS11021, CGS7184 and NS1619 and blocked by paxilline (Pax), iberiotoxin (IbTx) and charybdotoxin (ChTx) [27]. The BK_Ca_ channel is formed by four α pore-forming subunits that are encoded by a single *KCNMA1* gene (Slo1) [28,29]. The channels achieve functional diversity mainly through alternative splicing of the Slo1 mRNA and modulation by β- and γ-auxiliary subunits [30]. The mechanosensitivity of mitoBK_Ca_ channels has also been described in glioma cells [31]. Currently, four types of β subunits (β1–4) have been defined. According to the type of β subunit present in the tissue, the channel exhibits a distinct tissue-specific expression pattern and uniquely modifies the gating properties and sensitivity to channel modulators [32].

In this study, we have shown for the first time the functional properties of large-conductance Ca^2+^-regulated potassium channels in the inner mitochondrial membrane of a human bronchial epithelial cell line. Using the patch-clamp technique, we described two populations of mitoBK_Ca_ channels, one with control activity for which the probability of opening (*Po*) was approximately 50% at −40 mV and the second with high activity for which *Po* was approximately 80% at the same holding potential. Channel activity was sensitive to Ca^2+^ and to the BK_Ca_ channel opener NS11021. The change in channel activity modified mitochondrial respiration and the membrane potential. The molecular identity of the channel was characterized with RT-PCR, qPCR experiments and labeling with anti-BK_Ca_ channel subunit-specific antibodies.

## 2. Results

### 2.1. Identification of Single-Channel Activity in Mitochondria from Human Bronchial Epithelial Cells

We applied the patch-clamp technique to identify the single-channel activity in the inner mitochondrial membrane of human bronchial epithelial cells. We observed two types of large-conductance channel activities with kinetics similar to previously described mitoBK_Ca_ channels (Figure 1b). The probability of channel opening (*Po*) of the first type of channel increased from approximately 18% at −60 mV to 95% at positive voltages (Figure 1b). The second activity of the channel was characterized by a high opening probability at all tested holding potentials. *Po* increased from approximately 72% at negative potentials to approximately 98% at +60 mV. Both results were obtained in the presence of 100 µM Ca^2+^ (Figure 1b). A comparison of the opening probabilities of both channel activities is shown in Figure 2b. Experimental points for control activity presented in Figure 2b were fitted using a Boltzmann function. Based on fitting results, we calculated value of potential at which open probability of the channel is halfway (V_1/2_) was −38.4 mV. The mitoBK_Ca_ channel with control activity (first type) was recorded twice as often as (*n* = 32) the mitoBK_Ca_ channel with high activity (second type, *n* = 18) (Figure 1c). Both channel activities were observed in the same mitochondrial isolations. Importantly, no significant differences in the conductance of the channels with different activities were observed. The channel conductance with control activity was estimated to be 286.2 ± 5.5 pS and the conductance of the channel with high activity was 287.8 ± 2.8 pS. Both conductances were calculated based on the current-voltage relationship presented in Figure 2a. Rectification of the currents was not observed. The distribution of closed and open dwell times was also analyzed for both channel activities. The closing times of the channel with high activity are very low and close to 1 ms (Figure 2d), regardless of the potential. The channel with control activity has a pattern of closing times similar to the mitoBK_Ca_ channel from other tissues [22]. The mean time of closure is the highest at the −60 mV potential and is approximately 15 ms; then, it decreases to approximately 2 ms at the +60 mV potential (Figure 2c). Correspondingly, the open dwell times from −60 to +60 mV increased from approximately 4 to 30 ms for both channel activities. In both types of channel, the average open time also increased with increasing potential, and this relationship is characteristic of the mitoBK_Ca_ channel [21,32].

### 2.2. Pharmacological Properties of the Mitochondrial Potassium Channel from Human Bronchial Epithelial Cells

Substances known to modulate BK_Ca_ channel activity in a specific manner were used to examine the ion channel properties observed in the experiments described above. First, patch-clamp experiments were performed to test channel regulation by Ca^2+^. For these purposes, the calcium ion concentration was reduced from 100 µM (control conditions) to 1 µM. Representative single-channel recordings at different calcium ion concentrations for both types of observed channels (control and high activity channels) are presented in Figure 3. These experiments were performed at a holding potential of +40 mV. An immediate decrease in the open probability of BK_Ca_ channel was observed when a buffer with a low concentration of Ca^2+^ was applied. Differences between the various channel activities were observed when 30 and 10 µM Ca^2+^ concentrations were applied. We found that in the presence of 30 μM Ca^2+^ the channel with high activity remained active, whereas the open probability of the channels from the second group was reduced. Almost a complete reduction in open probability of both groups was observed at 1 µM Ca^2+^ (Figure 3b). The *Po* of channels with control and high activities was reduced from approximately 96% to 1% (Figure 3c,d). Additionally, based on fitting results in the presence of 50 μM Ca^2+^, calculated values of potential at which open probability of the channel is halfway (V_1/2_) was −5.2 mV and −16.5 mV for control and high activity, respectively.

Figure 4 illustrates the activity of the channel under control conditions and upon the addition of various concentrations of potassium channel opener NS11021, washout and application of 3 µM NS11021 plus 1 µM paxilline, a known inhibitor of the BK_Ca_ channel. This experimental protocol was used for the channel with control activity and the channel with high activity (Figure 4a,b). We observed that the addition of the NS11011 opener increased channel opening in a dose dependent manner. Application of paxilline plus NS11021 resulted in reduced *Po* of the channel starting with *Po* = 71% for the channel with control activity and *Po* = 81% for the channel with high activity, to 0% for both types of channel (Figure 4c,d). Furthermore, the addition of 5 µM NS1619, another BK_Ca_ channel activator, did not affect channel activity (data not shown). We also tested the effects of iberiotoxin, a BK_Ca_ channel inhibitor that binds to the external vestibule of the BK_Ca_ channel [21,33]. We performed a series of experiments on the channel with control activity. Iberiotoxin was administered from the intramembrane side (Figure 5b) and from the matrix side (Figure 5a). The addition of 300 nM Iberiotoxin from the intramembrane side resulted in the reduced *Po* of the channel after 17 min. At a potential of −40 mV, the probability of channel opening was reduced from 55% to 0%, and at a potential of +40 mV, it was reduced from 92% to 0% (Figure 5b). A decrease in open probability of the channel was not observed after the addition of 300 nM iberiotoxin from the matrix side. After 17 min of perfusion at a potential of −40 mV, the probability of channel opening was slightly inhibited from 65% to 49%, and at a potential of +40 mV it was slightly inhibited from 94% to 66% (Figure 5a).

### 2.3. Analysis of BK_Ca_ Channel Subunit Expression and Identification of BK_Ca_ Channel Proteins in Isolated Human Bronchial Epithelial Mitochondria

The expression of transcripts encoding the pore-forming α-subunit of the BK_Ca_ channel was detected in human bronchial epithelial cells (Figure 6a). We observed a single band of 167 bp indicating that the *KCNMA1* gene was expressed. Moreover, the expression of the regulatory subunits of the BK_Ca_ channel β1–β4 (*KCNMB1*, *KCNMB2*, *KCNMB3* and *KCNMB4*) was analyzed. Transcripts encoding β3 (band of 178 bp) and β4 (band of 149 bp) subunits were present in the tested samples. The obtained results correspond to the quantitative PCR experiment. The transcripts of the *KCNMA1*, *KCNMB3* and *KCNMB4* genes were detected in the early cycles of qPCR, whereas KCNMB1 and KCNMB2 were detected in the very late cycles of the reaction (cycle numbers > 34, which is close to the background value) (Figure 6b).

Western blotting with mitochondrial fractions was performed to confirm the presence of the BK_Ca_ channel protein in mitochondria (Figure 6c). Antibodies raised against the mammalian plasma membrane pore-forming α subunit and regulatory β3 and β4 subunits of the channel were used. In the human bronchial epithelial cells, two fractions of mitochondria (20 and 40 µg of protein sample) and the homogenate (20 µg of protein sample), a protein band of ~120 kDa was detected using an anti-K_Ca_1.1 antibody (Figure 6c). Furthermore, the anti-sloβ3 and anti-sloβ4 antibodies showed the presence of protein bands of ~27 kDa and ~24 kDa in the mitochondrial fractions (Figure 6c). A comparison of the protein levels between the homogenate and the mitochondrial fractions indicated a relatively high level of BK_Ca_ subunits in the mitochondria. The densitometry analysis of Western blots is presented in Figure 6d. Moreover, a correlation was observed between the increase in the signal of the tested subunits of the channel and the increase in the signal of the mitochondrial marker (subunit IV of cytochrome c oxidase) (Figure 6c).

### 2.4. Regulation of Mitochondrial Respiration and Membrane Potential by Potassium Channel Openers and Inhibitors

The mitochondrial nonphosphorylating respiratory rate was measured in permeabilized HBE cells cultured in potassium-containing calcium-free medium to study the effect of mitoBK_Ca_ channel modulators. Measurements were performed in the presence of 5 mM succinate (respiratory substrate for complex II), 0.5 µM rotenone (complex I inhibitor), 2.5 µM oligomycin (ATP synthase inhibitor) and 0.15 mM ATP.

Cells were permeabilized with digitonin. The addition of 1 µM NS11021 significantly increased the respiratory rate up to two-fold compared to the control. The higher the concentration of BK_Ca_ channel activator applied, the higher the respiration rate (Figure 7a). In the presence of the channel inhibitor paxilline, the addition of NS11021 increased respiration to a much lesser extent.

The mitochondrial membrane potential of HBE cells was analyzed with JC-10 dye, which forms red J-aggregates in mitochondria with a high membrane potential but remains a green monomer in cells that have lost the mitochondrial membrane potential. The addition of 1, 3, and 5 µM NS11021 depolarized the mitochondrial membrane potential (∆Ψ) in a dose-dependent manner, while the incubation of cells with 1 µM paxilline in the presence of 3 µM NS11021 reduced this effect (Figure 7b). Treatment with 100 and 300 nM FCCP (mitochondrial uncoupler) was used as a positive control for maximal depolarization of mitochondria. The scatter plots show that the majority of potassium channel opener NS11021-treated cells shifted towards JC-10 green fluorescence compared to controls (Figure 7c).

## 3. Discussion

Several types of potassium channels (mitoK) have been detected in the inner mitochondrial membranes, including large-conductance Ca^2+^-regulated potassium (mitoBK_Ca_) channels [5,21,22,34]. In general, both the biophysical and pharmacological properties of mitoK channels are similar to the properties of the channels present in the plasma membrane. Because the channels are located in the inner mitochondrial membrane, their function is quite unique and complex. Mitochondria are cellular hubs for a variety of biochemical processes; hence, a complete description of the functional role of mitoK channels is still needed. Additionally, very little is known about the transport of potassium through the inner mitochondrial membrane of epithelial cells.

The discovery of the mitoBK_Ca_ channel in the inner mitochondrial membrane revealed an additional role of mitoK channels [15]. Further studies have shown that activation of the mitoK channel by potassium channel openers induces protective mechanisms, e.g., in cardiac myocytes [10]. This finding initiated the search for mitoK channels in other organs and for their cytoprotective role [35]. K^+^ flux through the inner mitochondrial membrane affects the coupling between electron transport and ATP synthesis, which is necessary for maintaining the cellular energy balance. Activation of the mitoK channel appears to be an important mechanism to control the integrity of the inner mitochondrial membrane and regulate ROS synthesis by inducing a slight decrease in the mitochondrial membrane potential [6,26].

Here, for the first time, we described the electrophysiological, pharmacological and functional properties of the mitoBK_Ca_ channel in human bronchial epithelial mitochondria. We detected two types of channel activities with similar conductances (286.2 ± 5.5 pS and 287.8 ± 2.8 pS) but different control *Po*. For this reason, we distinguished two populations of the channel. The first as a channel with a control activity which was recorded more often and the second channel with high activity appearing half as often. Interestingly, the presence of two different mitoBK_Ca_ channel populations in the inner mitochondrial membrane of the brain has already been observed, and the channels showed variations in gating and ATP sensitivity [29]. In cardiac tissue, two populations of mitoBK_Ca_ were also described. Both groups had a very similar conductance, and the main difference between them was their sensitivity to calcium ions [36]. Both channel populations observed in the present study were voltage-dependent, with higher open-state probability at positive potentials, and were sensitive to Ca^2+^, the potassium channel opener NS11021 and the channel blocker paxilline. These electrophysiological properties are similar to activities previously described for mitoBK_Ca_ channels in mammalian mitochondria from cardiac [19], brain [14,16], skeletal muscle [17], endothelial [20] and dermal fibroblasts [21].

The sensitivity of the BK_Ca_ channel to calcium ions is due to the presence of a Ca^2+^-sensing domain located at the C-terminus of the α subunit. The soluble C-terminus, which occupies two-thirds of the whole protein, contains two regions that sense Ca^2+^, known as K^+^ conductivity regulators (RCK) 1 and 2, which together are sufficient to activate BK_Ca_ at physiological Ca^2+^ concentrations [37]. A decrease in the probability of opening in both observed types of mitoBK_Ca_ channels in the presence of low calcium concentrations was observed. This decrease in the opening probability is a canonical property of mitoBK_Ca_ channels and was previously described in channels from various tissues, including fibroblasts [22] and the endothelium [21].

We also observed the activation of both channel populations by a specific BK_Ca_ channel opener, NS11021 [38]. The activation of the channels with control activity was much greater than that of the second type of channel. The initial *Po* of the high activity channel is high; therefore, the activation by NS11021 was small and caused the channel to be almost completely open (*Po* ~ 97% at −40 mV, after treatment with 5 µM NS11021). It has been shown that NS11021 activation of BK_Ca_ channel is largely due to the stabilization of an open state at the pore gate domain (PGD) [39,40]. It is possible, therefore, that a similar mechanism is responsible for the permanently higher activity of this fraction of channels. Interestingly, this channel behavior could be due to mitochondrial-specific factors. However, this needs more detailed study. Another valuable tool for studying the mitochondrial large-conductance Ca^2+^-regulated potassium channel is channel blockers. In the present study, we used the diterpene paxilline and peptide iberiotoxin. We observed that both types of mitoBK_Ca_ channels were inhibited by 1 µM paxilline. Previous studies revealed that iberiotoxin, the second inhibitor used in our study, shows high affinity for BK_Ca_ channels when applied to the extracellular side of the channel [41]. We conducted experiments in two conformations: in one conformation, the inhibitor was administered from the side of the mitochondrial matrix, and in the other conformation, the inhibitor was administered from the side of the intermembrane space. The application of the drug from the intermembrane space resulted in a decrease in the mitoBK_Ca_ channel open probability. This observation suggests a topology of the channel with the C-terminus in the matrix, as was observed in other cell types, such as heart or glioma cells [20,42]. However, we observed a slightly reduced *Po* of the channel when the inhibitor was administered from the matrix side. This result was also observed previously in U-87 MG cells [34]. This finding suggests either diffusion of the toxin across the membrane or an alternative membrane-related mechanism of the channel inhibition. Moreover, ROMK channels and mitoSK_Ca_ channels are potentially blocked by peptide inhibitors applied on the opposite side of the channel pore [43,44]. This suggests that peptide inhibitors may interact with potassium channels in several ways. However, this hypothesis requires more in-depth research. These pharmacological properties proved that the mitochondrial large-conductance Ca^2+^-regulated potassium channel is present in human bronchial epithelial cells.

The molecular composition of the BK_Ca_ channel differs depending on the tissue [45]. The channel is formed by four α pore-forming subunits that are encoded by a single Slo1 gene [28]. The BK_Ca_ channels achieve functional diversity mainly due to alternative splicing of the Slo1 mRNA and modulation by additional β subunits [46]. Four types of β subunits (β1–4) have been identified, and each type exhibits a distinct tissue-specific expression pattern and uniquely modifies the gating properties of the channel [32]. In the present study, the mRNA expression of the α, β3 and β4 subunits of the BK_Ca_ channel in human bronchial epithelial cells was detected. These results were confirmed by immunodetection of mitoBK_Ca_ channel proteins in the mitochondria and homogenate of HBE cells using antibodies raised against the BK_Ca_1.1 α subunit and β3-4 subunits. In addition, the β1 subunit has been shown to be an auxiliary subunit of mitoBK_Ca_ in cardiac mitochondria [36] and β3 in mitoBK_Ca_ from human fibroblasts [22]. On the other hand, electrophysiological studies confirmed the presence of two populations of mitoBK_Ca_ (control and high activity) channels in HBE cells. These populations correspond to the possible regulation of α pore formation by β3 and β4 of the BK_Ca_ channel auxiliary subunits. Based on the iberiotoxin sensitivity, we suggest that mitoBK_Ca_ regulated predominantly by β3 is present in the mitochondria of the HBE cell model. This variant is regulated by IbTx, and the β4-complexed BK channel is known to be resistant to both IbTx and ChTx [45,47].

Induction of potassium ion flux via the mitoBK_Ca_ channel should interfere with mitochondrial respiration and mitochondrial ∆Ψ [21,22]. In the present study, we examined the effects of the BK_Ca_ channel opener NS11021 on ΔΨ and mitochondrial respiration in energized mitochondria. In the micromolar concentration range, NS11021 modulated the respiratory rate under nonphosphorylating conditions. This effect was diminished in the presence of the mitoBK_Ca_ channel inhibitor paxilline. The effect obtained corresponds to the results obtained in the EA.hy962 cell line, where the two BK_Ca_ channel activators NS1619 and NS11021 were compared. Micromolar concentrations of NS11021 induced respiration in the presence of an iberiotoxin on isolated endothelial mitochondria [21]. In contrast, micromolar concentrations of NS11021 not only modulated the respiration rate but also induced depolarization of ΔΨ in a dose-dependent manner, which was partially reversed by a mitoBK_Ca_ channel blocker. MitoBK_Ca_ channel activators, both synthetic (NS1619 and NS11021) and natural (flavonoid derivative-naringenin) potassium channel openers, increase respiration and induce the depolarization of ΔΨ of mitochondria isolated from various tissues and cell lines, e.g., skeletal muscle [18], brain [17], endothelial cells or skin fibroblasts [21,22]. These observations have prompted the hypothesis that potassium channel activation leads to the influx of potassium cations into the mitochondrial matrix, increases respiration and causes mitochondrial depolarization. Taken together, the present findings support the hypothesis that selective activation of mitochondrial BK_Ca_ channels modulates K^+^ uptake and induced depolarization ΔΨ [48]. In addition, activation of the mitoBK_Ca_ channel regulates mitochondrial reactive oxygen species synthesis [14]. These properties are likely required to protect cells during recovery from metabolic stress [49].

Large-conductance Ca^2+^-regulated potassium channels located in the plasma membrane and in the inner mitochondrial membrane have become a potential tool to protect cells from death by modulating intracellular calcium signaling. A significant effect of the activation of mitochondrial potassium channels, including mitoBK_Ca_ channels, is the attenuation of mitochondrial reactive oxygen species under oxidative stress, ensuring cell protection [50,51].

## 4. Materials and Methods

### 4.1. Cell Culture

Immortalized human bronchial epithelial cells (16HBE14o-) were obtained from Dieter Gruenert from the University of California (San Francisco, CA, USA) (UC Case No. SF1992-A66) [2,52]. Human bronchial epithelial (HBE) cells were cultured in MEM (Sigma-Aldrich, St. Louis, MO, USA) supplemented with 10% FBS (Gibco, Carlsbad, CA, USA), 100 U/mL penicillin (Sigma-Aldrich, St. Louis, MO, USA) and 100 µg/mL streptomycin (Sigma-Aldrich, St. Louis, MO, USA) at 37 °C in a humidified atmosphere with 5% CO_2_ [2,52]. The cells were fed and reseeded when they reached approximately 90–100% confluence (usually every third day).

### 4.2. Mitochondrial and Mitoplast Preparation

Mitochondria were prepared from a human bronchial epithelial (HBE) cell line. 16HBE14o- cells from two culture flasks were collected in phosphate-buffered saline (PBS) and centrifuged at 400× *g* for 8 min. The cell pellet was resuspended and homogenized (Wheaton homogenizer, Mainz, Germany) in a preparation solution (250 mM sucrose, 5 mM HEPES, and 1 mM EGTA, pH = 7.2). The homogenate was centrifuged at 9200× *g* for 10 min in 4 °C. Next, the pellet was suspended and centrifuged at 750× *g* (10 min, 4 °C). The supernatant was transferred to a new tube and centrifuged at 9200× *g* for 10 min at 4 °C. The pelleted mitochondria were resuspended in approximately 0.3 mL of preparation solution. All procedures were performed on ice at 4 °C. For electrophysiological measurements, mitoplasts were prepared by incubating the mitochondria in a hypotonic solution composed of 5 mM HEPES and 100 µM CaCl_2_, pH 7.2, for approximately 1–2 min to induce swelling and breakage of the mitochondrial outer membrane. Then, a hypertonic solution (750 mM KCl, 30 mM HEPES and 100 µM CaCl_2_, pH 7.2) was added to inhibit swelling of the mitochondria, and isotonic medium with suspended mitoplasts was finally obtained.

### 4.3. Patch-Clamp of the Inner Mitochondrial Membrane

Patch-clamp inside-out experiments using mitoplasts were performed as described previously [9,53,54]. A pipette of borosilicate glass (Harvard Apparatus GC150-10, Holliston, MA, USA) with a resistance of 15 to 20 MΩ was filled with isotonic solution (150 mM KCl, 10 mM HEPES and 100 µM CaCl_2_, pH 7.2). Mitoplasts prepared as described above were recognized under the microscope by observing the characteristic round shape, size, transparency and occurrence of the characteristic “cap”, which is formed by remnants of the outer mitochondrial membrane. The control solution was an isotonic solution containing 100 µM Ca^2+^, and all BK_Ca_ channel modulators (NS11021, NS1619, paxilline and iberiotoxin) were added as dilutions in an isotonic solution. The solutions were prepared with different concentrations of calcium ions 1, 10, 30 µM (150 mM KCl, 10 mM HEPES and 1 mM EGTA), 50 and 70 µM (150 mM KCl, 10 mM HEPES and 2 mM EGTA) and the corresponding concentration of CaCl_2_ at pH 7.2. We used a flow system driven by a peristaltic pump to apply the test solution from the test tube to the vessel in which the measuring pipette was located (scheme shown in Figure 5a). The substances NS11021, NS1619 and paxilline were added to the matrix side. One set of experiments involved the administration of the iberiotoxin from the intermembrane space side. The tip of the pipette was filled with isotonic solution by negative pressure, and the rest of the pipette was filled from the back with solution containing the isotonic solution with 300 nM iberiotoxin (IbTx). Over time, the test substance reached the ion channel by diffusion (scheme shown in Figure 5b). Voltages applied to the patch-clamp pipette interior are reported. Single-channel recording was performed using a patch-clamp amplifier (Axopatch 200B, Molecular Devices Corporation, San Jose, CA, USA). The recordings were low-pass filtered at 1 kHz, sampled at a frequency of 100 kHz and acquired by Clampex 10.7 software (Molecular Devices Corporation, San Jose, CA, USA). The traces of the experiments were recorded in a single-channel mode. The presented channel recordings are representative of the most frequently observed conductance for the indicated condition. The conductance of the mitoBK_Ca_ channel was calculated from the current–voltage relationship (Figure 2a). The probability of channel opening (*Po*) and the mean time of closure/opening were determined using the single-channel search mode of the Clampfit 10.7 software.

### 4.4. Identification of BK_Ca_ Channel Subunit Transcripts

HBE cells from three 75 cm^2^ cell culture flasks at 90–100% confluence were washed twice with PBS, scraped in PBS and centrifuged at 400× *g* for 8 min. Total RNA was extracted using the RNeasy Mini Kit according to the manufacturer’s protocol (Qiagen, Hilden, Germany). During isolation, DNase digestion (RNase-Free DNase Set; Qiagen, Hilden, Germany) was performed in an additional step. The RNA concentration was measured using a standard technique for concentration and purity measurements—UV absorbance measured with a spectrophotometer. First-strand cDNA synthesis from RNA templates was performed according to the protocol RevertAid First Strand cDNA Synthesis Kit (Thermo Fisher Scientific, Waltham, MA, USA). The resulting cDNAs were used as templates for standard qualitative PCR (RedTaq DNA polymerase, Sigma-AldrichSt. Louis, MO, USA) or quantitative polymerase chain reaction (qPCR) using Power SYBR Green PCR Master Mix (Thermo Fisher Scientific, Waltham, MA, USA). Real-time PCR was performed in triplicate, and reactions were run for each sample in 96-well plates with first-strand cDNAs from four different isolates as the template. PCR was conducted with the following primers: *KCNMA1* forward primer: 5′CCGCAGACACTGGCCAATAG 3′, reverse primer: 5′GAGCATCTCTCAGCCGGTAA 3′ (predicted product size—167 bp); *KCNMB1* forward primer: 5′CCAGAACCAGCAGTGCTCCTACAT 3′, reverse primer: 5′GCTCTTGGAATTTGGCTCTGAC 3′ (predicted product size—95 bp); *KCNMB2* forward primer: 5′CACACTCCTGCGCTCATAC 3′, reverse primer: 5′ACCTGGAGGCAGGGGTAC 3′ (predicted product size—147 bp); *KCNMB3* forward primer: 5′ATATCATGGACGACTGGCTG 3′, reverse primer: 5′CTATCTTGGTGGCACTTAGG3′ (predicted product size—178 bp); and *KCNMB4* forward primer: 5′GTTCGAGTGCACCTTCACCT 3′, reverse primer: 5′ AGGAGCACTTGGGGTTGGT3′ (predicted product size—169 bp). The PCR products were separated by electrophoresis on a 2% agarose gel and visualized with Midori Green (Nippon Genetics Europe GmbH). Both qualitative and quantitative PCRs were performed using four independent cDNA preparations.

### 4.5. SDS-PAGE and Immunoblotting

Mitochondria were isolated from HBE cells using differential centrifugation and purification as described in the protocol below. The cells from ten 150 cm^2^ cell culture dishes at 90–100% confluence were washed twice with PBS, trypsinized and centrifuged at 600× *g* for 15 min. The cell pellet was resuspended in ice-cold isolation buffer (210 mM mannitol, 70 mM sucrose, 0.5 M HEPES-KOH, pH 7.2 and 2.5 mg/mL BSA) supplemented with 200 μg/mL digitonin. After reaching 80–90% permeabilized cells, homogenization (Wheaton homogenizer, Mainz, Germany, USA) was performed. The homogenate was centrifuged twice at 600× *g* for 5 min at 4 °C. The supernatant was transferred to a new tube and centrifuged at 10,000× *g* for 1 h at 4 °C. Next, the pellet was suspended in isolation buffer and centrifuged again at 10,000× *g* (30 min, 4 °C). The mitochondrial pellet fraction was suspended in isolation buffer without BSA and centrifuged twice at 10,000× *g* for 15 min at 4 °C. For Western blot analysis, 20 and 40 μg of protein were separated on 4 or 12.5% SDS-PAGE gels and transferred to PVDF membranes (Bio-Rad, Hercules, CA, USA). The membranes were blocked with 10% skim milk in TBST (50 mM TRIS, 150 mM NaCl, and 0,2% Tween-20, pH = 8.4) for 1 h at room temperature. Primary anti-KCa 1.1 (NeuroMab—L6/60), anti-sloβ3 (NeuroMab—N40B/18), anti-sloβ4 (Alomone Labs—APC061) and anti-COX IV (Cell Signaling—4844C) antibodies were diluted in 5% skim milk in TBST buffer and incubated overnight at 4 °C. Proteins were detected with appropriate secondary antibodies conjugated to horseradish peroxidase (prepared in 5% skim milk in TBST) and incubated for 1 h at room temperature followed by an enhanced chemiluminescence system (GE Healthcare).

### 4.6. Measurements of Oxygen Consumption

Human bronchial epithelial cells from one 75 cm^2^ cell culture flask at 90–100% confluence were washed twice with PBS, a trypsin solution was added and incubated for 5 min at 37 °C in an incubator, and MEM containing 10% FBS was subsequently added. The resuspended cells were transferred to a centrifuge tube and centrifuged at 400× *g* for 6 min. The cell pellet was suspended in respiration buffer (0.5 mM EGTA, 3 mM MgCl_2_·6 H_2_O, 60 mM lactobionic acid, 20 mM taurine, 10 mM KH_2_PO_4_, 20 mM HEPES, 110 mM D-sucrose and 1 g/L BSA, pH = 7.1). 16HBE14o- cells were counted using a cell counter and resuspended in respiration buffer to a final density of 1.5 × 10^6^ cells/mL. Oxygraph-2k (O2k, Oroboros Instruments, Innsbruck, Austria) was used for measurements of respiration [55]. Up to four O2k instruments (two chambers) were used in parallel. Cells were permeabilized with 10 µg/mL digitonin. Measurements were performed in the presence of 5 mM succinate (as a respiratory substrate) with 0.5 µM complex I inhibitor rotenone, 2.5 µM oligomycin (to inhibit ATP synthase) and 0.15 mM ADP. All experiments were performed at 37 °C.

### 4.7. Measurements of the Mitochondrial Membrane Potential

The JC-10 biphasic cationic dye is able to infiltrate both the cell and mitochondrial membranes in the form of a monomer with green emission (525 nm). When the mitochondrial membrane potential is elevated, the JC-10 dye forms J aggregates with orange emission (590 nm). This property allowed us to determine the proportions of cells that contained mitochondria with higher or lower mitochondrial membrane potentials. Human bronchial epithelial cells were incubated with the appropriate BK_Ca_ channel modulators. Cells treated with various substances were harvested at specified time points and centrifuged at 400× *g* for 6 min. The HBE cell pellet was suspended in 1 mL of MEM and counted using a cell counter. Then, 500,000 cells were resuspended in 500 μL of JC-10 dye-loading solution from a JC-10 mitochondrial membrane potential assay kit (Abcam, Cambridg, UK). The dye-loaded cells were incubated in a 37 °C, 5% CO_2_ incubator for 30 min in the dark. Fluorescence was subsequently recorded using a flow cytometer (BD LSRFortessa™, Franklin Lakes, NJ, USA) at an excitation wavelength of 490 nm and emission wavelengths of 520 nm and 590 nm. Experiments with JC-10 assays were performed at the Laboratory of Cytometry, Nencki Institute of Experimental Biology, Polish Academy of Sciences, Warsaw, Poland.

### 4.8. Statistical Analysis

The results are reported as the means ± SD obtained from at least three independent experiments. An unpaired two-tailed Student’s *t*-test was used to identify any significant differences. Differences were considered significant if *p* < 0.05 (*), *p* < 0.01 (**), *p* < 0.001 (***) or *p* < 0.0001 (****).

## 5. Conclusions

In summary, in the present study, we identified and characterized the mitoBK_Ca_ channel in the inner mitochondrial membrane of human bronchial epithelial cells. Mitochondrial localization of this channel was confirmed using patch-clamp experiments and biochemical and molecular biology tools. Two populations of the BK_Ca_-type channel were detected. MitoBK_Ca_ regulated by β3 is predominantly present in the mitochondria of HBE cells. Additionally, we examined the topology of the channel in the inner mitochondrial membrane and found that the C-terminus of the protein is localized to the mitochondrial matrix. We are convinced that our findings provide an opportunity to identify a new strategy for cytoprotection that includes the participation of mitochondrial potassium channels in an epithelial model.

## Figures and Tables

**Figure 1 molecules-26-03233-f001:**
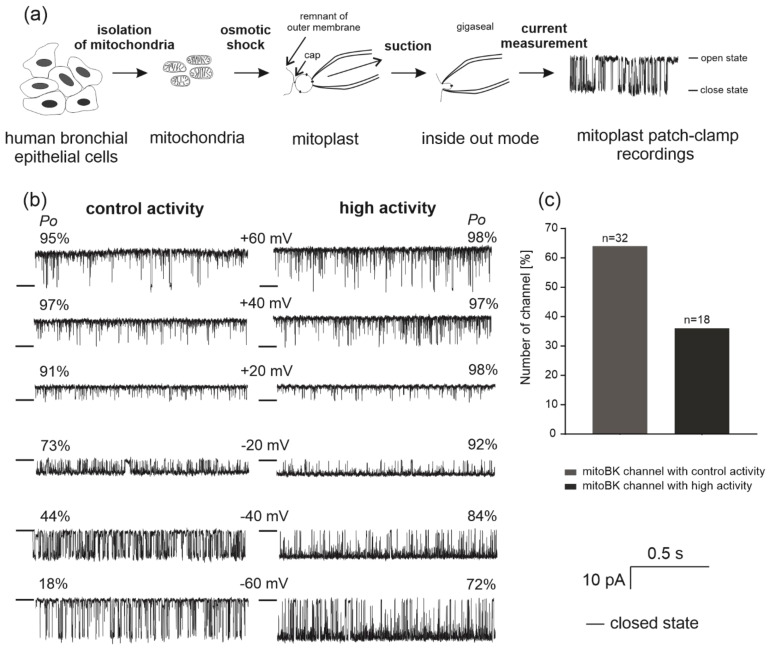
Outline of the mitochondrial patch-clamp experiments and comparison of the two types of recorded channel activities. (**a**) Scheme of the preparation of human bronchial epithelial mitochondria, mitoplasts, mitoplast patching, patch-clamp inside out mode and single-channel recordings. (**b**) Comparison of the two types of single-channel current-time recordings of the mitoBK_Ca_ channel: control and high activity in a symmetric 150/150 mM KCl isotonic solution (in the presence of 100 µM Ca^2+^) at different voltages (*n* = 10). “—” indicates a closed channel state. “*Po*” represents open probability analysis of single channel recordings. (**c**) Comparison of the numbers of mitoBK_Ca_ channels observed with control (*n* = 32) and high activity (*n* = 18) in the analyzed patches.

**Figure 2 molecules-26-03233-f002:**
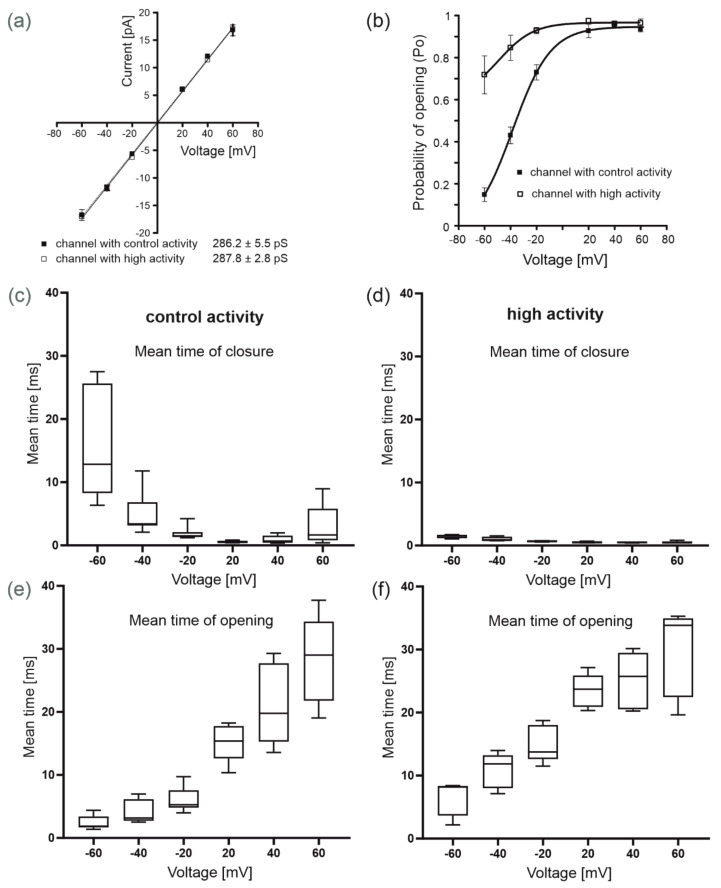
Biophysical properties of both mitoBK_Ca_ channels (channels with control and high activity) present in the inner mitochondrial membranes of human bronchial epithelial cells. (**a**) The current-voltage relationship based on single-channel recordings in a symmetric 150/150 mM KCl isotonic solution (100 µM Ca^2+^) of the channel with control and high activity (*n* = 4). (**b**) Analysis of the open probability of the mitoBK_Ca_ channels with control and high activities in the presence of 100 µM Ca^2+^ at different voltages (*n* = 4). (**c**,**e**) Distribution of the mean times of closure and opening of the mitoBK_Ca_ channel with control activity (*n* = 4). (**d**,**f**) Distribution of the mean times of closure and opening of the mitoBK_Ca_ channel with high activity (*n* = 3). Notes: (**a**,**b**) data are presented as the means ± SD under control conditions (100 µM Ca^2+^); (**c**–**f**) the boxes include values for the mean time, the line across the box indicates the median, and whiskers show the minimum and maximum values for mean time of closure/opening.

**Figure 3 molecules-26-03233-f003:**
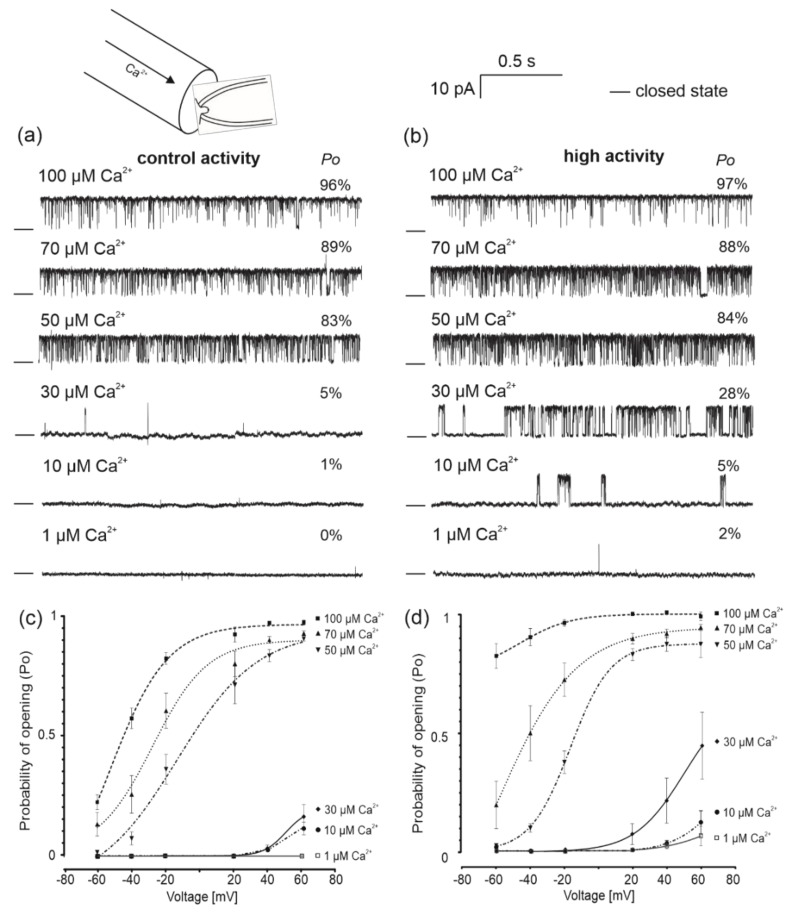
Regulation of the two populations of mitoBK_Ca_ channels by calcium ions. (**a**,**b**) Representative single-channel current-time recordings of the mitoBK_Ca_ channel in the presence of different calcium concentrations (1, 10, 30, 50, 70 and 100 μM Ca^2+^). Both types of channel activities recorded in 150/150 mM KCl isotonic solution at +40 mV are presented (*n* = 3). “—” indicates a closed channel state. “*Po*” represents the average channel open probability. (**c**,**d**) Quantification of the open probability of channels in the presence of 100, 70, 50, 30, 10 and 1 µM Ca^2+^. Data are presented as the means ± SD (*n* = 3).

**Figure 4 molecules-26-03233-f004:**
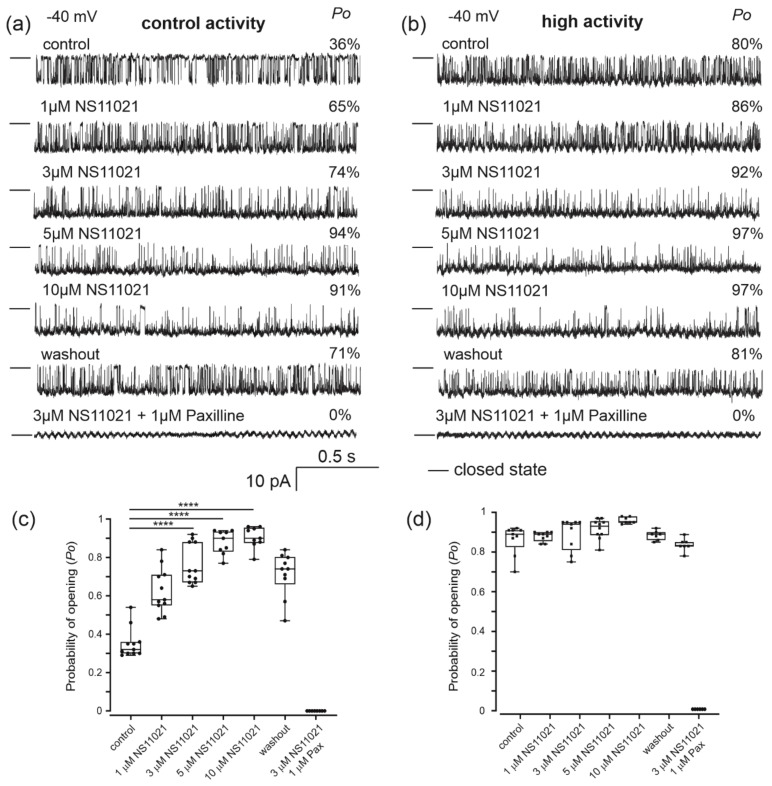
Effects of the potassium channel opener NS11021 on the BK_Ca_ channel present in human bronchial epithelial mitochondria. (**a**,**b**) Representative traces of single-channel recordings of mitoBK_Ca_ channels from human bronchial epithelial mitochondria in symmetric 150/150 mM KCl control solution (100 µM Ca^2+^) under different conditions. Sequences of the applied drugs (NS11021 and paxilline) are presented in the graph. Left panel, (**a**) mitoBK_Ca_ channel with control activity, and right panel, (**b**) mitoBK_Ca_ channel with high activity. Single-channel activities from different experiments (*n* = 3) recorded at −40 mV. “—” indicates a closed channel state. The *Po* analysis (%) of single channel events is shown. (**c**,**d**) Analysis of the mitoBK_Ca_ channel open probability under different conditions: control (100 µM Ca^2+^), different concentrations of NS11021, washout and 3 µM NS11021 plus 1 µM paxilline. Left panel, (**c**) mitoBK_Ca_ channel with control activity, and right panel, (**d**) mitoBK_Ca_ channel with high activity. Notes: the boxes include values for *Po*, the dots are single counted traces, the line across the box indicates the median, and whiskers show the minimum and maximum values for *Po*. The data are presented as the means ± SD (*n* = 3). *p* < 0.0001 (****).

**Figure 5 molecules-26-03233-f005:**
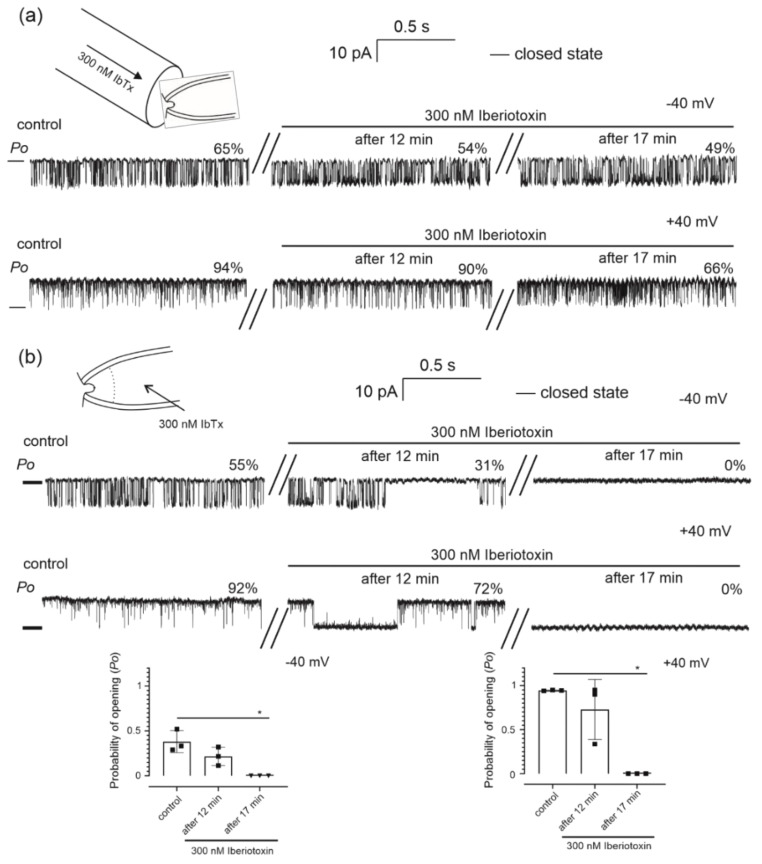
Effect of iberiotoxin on mitoBK_Ca_ channel activity. (**a**) Scheme of the mitochondrial matrix direction of iberiotoxin administration to the mitoBK_Ca_ channel with control activity. Representative single-channel current-time recordings of the mitoBK_Ca_ channel incubated with 300 nM iberiotoxin (*n* = 3) for different times at −40 mV and +40 mV. ‘—’ indicates a closed channel state. The *Po* analysis (%) of single channel events is shown. (**b**) Scheme of the intermembrane space direction of iberiotoxin administration to the mitoBK_Ca_ channel with control activity. Representative traces showing the effects of the time of incubation with 300 nM iberiotoxin (*n* = 3) recorded at −40 mV and +40 mV. “**—**” indicates a closed channel state, and the *Po* analysis (%) of single channel events is shown. The charts below show the quantification of the mitoBK_Ca_ channel open probability in the presence of the control (100 µM Ca^2+^) and 300 nM iberiotoxin after different incubation times. *p* < 0.05 (*).

**Figure 6 molecules-26-03233-f006:**
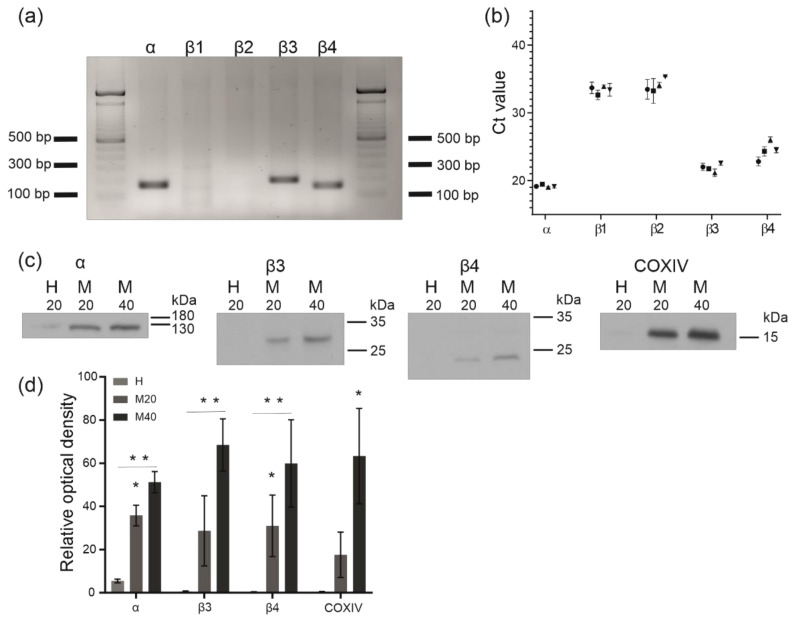
Analysis of gene expression and the immunodetection of the BK_Ca_ channel α pore forming and auxiliary β subunits. (**a**) PCR amplification of BK_Ca_ channel subunits. α, Product of the amplification of *KCNMA1* gene (*n* = 3); β1–4, products of the amplification of transcripts encoded by KCNMB1-KCNMB4 (*n* = 3). (**b**) qPCR detection of the BK_Ca_ channel subunits; Ct: cycle threshold; (*n* = 4). (**c**) Representative images of immunostaining for BK_Ca_ channel subunits. Western blot analysis of proteins from the cell homogenate (H; 20 µg of total protein) and mitochondrial fractions (M at two concentrations, 20 µg and 40 µg), loaded into each lane (*n* = 3). (**d**) Quantification of band intensities was performed using ImageJ software. The number of Western blots analyzed was three per group. *p* < 0.05 (*), *p* < 0.01 (**).

**Figure 7 molecules-26-03233-f007:**
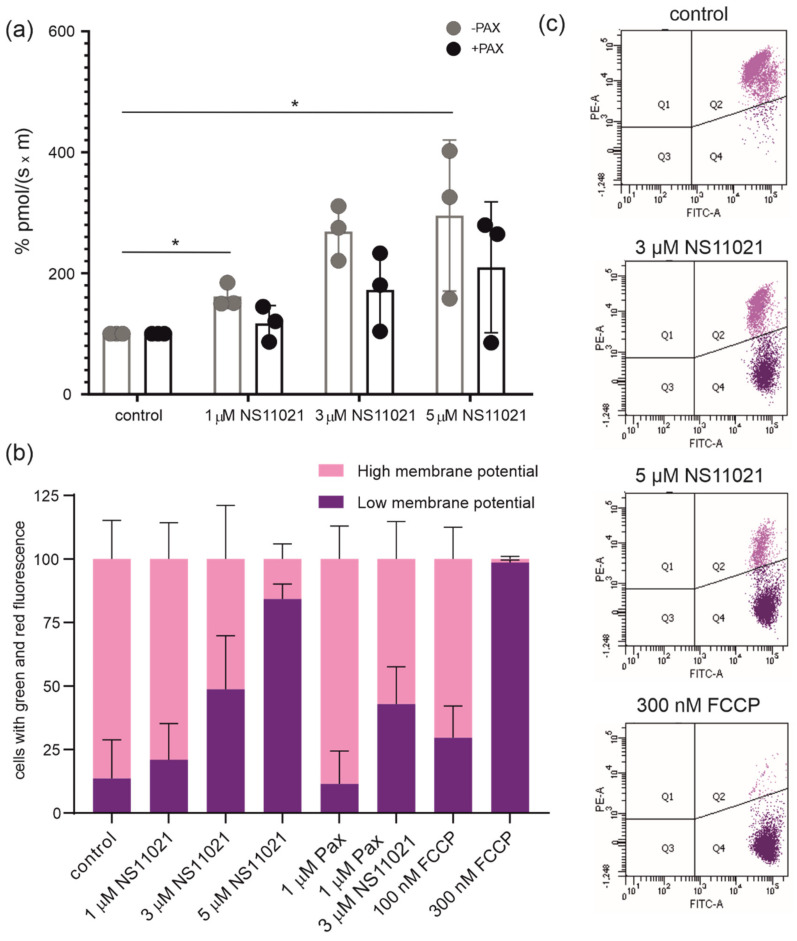
Effects of mitoBK_Ca_ channel modulators on the membrane potential and respiration rate. (**a**) Mitochondrial respiratory capacity; a final density of 1.5 × 10^6^ cells/mL was plated in each chamber and the NS-induced changes in respiratory rate in the presence (•) or absence (•) of 10 µM paxilline were detected. Data are reported as the means ± SD (*n* = 3), and each dot represents a separate experiment. (**b**) Measurement of the mitochondrial membrane potential using the JC-10 dye. Changes in red vs. green fluorescence were used to measure the mitochondrial membrane potential after treatment of 16HBE14o- cells with BK_Ca_ channel modulators. FCCP-treated cells were used as positive controls. (**c**) Representative dot plots of stained 16HBE14o- cells obtained using a flow cytometry analysis. Cells that appear in quadrant 4 represent cells with depolarized mitochondrial membranes. *p* < 0.05 (*).

## Data Availability

No additional data available.

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
