# Peer review of "Identification of the Large-Conductance Ca2+-Regulated Potassium Channel in Mitochondria of Human Bronchial Epithelial Cells"

_molecules, 2021, doi:10.3390/molecules26113233_

Round 1

Reviewer 1 Report

The authors provide evidence for two types of BK channels in recordings from mitoplasts. The authors are to be commended on the quality of the single channel records and the logical layout of the paper. I have a number of queries in relation to this study, which if addressed, I hope will strengthen the paper.

  1. EGTA is unlikely to be able to adequately buffer [Ca2+] above 300 nM (see Patton C, Thompson S, Epel D. Some precautions in using chelators to buffer metals in biological solutions. Cell Calcium. 2004 May;35(5):427-431. DOI: 10.1016/j.ceca.2003.10.006. ) Have the authors measured [Ca2+] with a Ca2+ If they use HEDTA to buffer Ca2+, do they see similar results?

  1. Fit data in Figure 2b with a Boltzmann and state the activation V1/2 in both

  1. Could the authors clarify if ‘control activity’ and ‘high activity’ channels were ever observed on the same day in the same dish? If so, could they state this in the results section. If they were not observed in the same dish, could the difference be due to variations in the quality of the mitoplasts obtained?

  1. The authors also need to examine if regulatory g and LINGO1 subunits could contribute to the ‘high activity’ BK channels. Regulatory gamma subunits have been found in airway epithelium (Manzanares et al., (2015). J Biol Chem 290: 25710–25716, 2015. doi:10.1074/jbc.M115.670885), so their presence may explain the high Po in the high activity channels.

  1. Authors suggest that the beta3 subunit may be present, yet no inactivating channels were recorded. Have the authors examined if inactivation occurs in mitochondrial BK channels. They will obviously need stepping protocols to observe this. If they don’t every observe inactivation, could they explain why, given that beta3 subunits appears to be present.

  1. L167 I assume you are referring to Fig 3c and 3d?

  1. L156 to L165 The authors use the terms inactivation, deactivation and inhibition incorrectly and this will confuse readers. E.g. “Immediate channel inactivation was observed when buffer…” Please change the wording in the offending sentences. Decreasing [Ca2+] doesn’t inhibit the channel, it simply reduces it Po.

  1. L192 to L195. The authors demonstrated that iberiotoxin completely blocked BK channels when applied to the intramembrane side, but also appeared to reduce Po (by ~30%) when applied from the matrix side. Is it possible that this effect is an artefact and could be due to rundown of channel activity? The inclusion of time dependent controls would be beneficial to answer this.

  1. Are the expected amplicons for BKalpha and BKbeta3 both 167bp (as stated in L208 and L210), or was this a mistake? The reason I ask is that the band in Figure 6A for BKb3 appears to be larger than that for BK.
  2. L308 to L310. Either show the data for the lack of effect of NS1619, or remove this sentence.

Author Response

Dear Reviewer,

Plaese find an enclsed responces in docx file.

Best regards,
Piotr Bednarczyk

Reviewer 2 Report

The manuscript reports a thorough and careful biophysical and biochemical investigation of the mitochondrial K(Ca)1.1/BK(Ca) channel found to be present in the inner membrane of a bronchial epithelial cell line. MitoBK(Ca) has been described in mitochondria from various other types of cells. This one presents some interesting elements of novelty concerning its kinetic behaviour in patch-clamp experiments and the molecular composition of its complex. I do have some questions and suggestions for improvement.

Is it possible to infer something about the high-activity variant of the channel from the effects of NS11021 on the control-activity variant? In other words, is it possible to attempt a comparison between the NS11021-treated control-activity channel and the untreated high-activity channel? Leads for discussion/speculation on this may come from the recent paper by Michael E. Rockman et al., 2020 (JGP) and the commentary by Jianmin Cui. Also, any speculation as to why NS1619 does not work?

Their findings induce the Authors to suggest that the control-activity channel may represent a complex containing the β3 subunit. This is likely to be correct, but then one wonders about the presence of non-negligible amounts of subunit β4 (Fig. 6). Is it possible that the high-activity channel represents a complex with β4? As the Authors write (lines 346-7) β4-comprising channels are resistant to IbTx. In Fig. 5 the manuscript shows inhibition of channel activity by IbTx, specifying that the results shown pertain to the control-activity channel. Have the Authors tested IbTx on the high-activity channel? If it turned out to be less sensitive to the toxin, this might support a tentative molecular identification and distinction between the two types.

The sentence on lines 126-127 (“This finding may be due to an increased probability of channel opening, which causes the channel to remain closed for a very short time”) is obvious (of course the probability of opening is high if the mean closed time is short) and might as well be omitted.

Lines 193 and 323-6. Is not a slow and local diffusion of the toxin across the patch a more economic hypothesis to explain the partial inhibition of the channel?

Line 367: “while maintaining ΔΨ”: ?. NS11021 depolarizes mitochondria, right?

Lines 374-375: “is the attenuation of mitochondrial regulation in cells under oxidative stress”. Unclear. Explain

Line 422: Axopatch not Axppatch

Line 423. 1 KHz seems a rather low filter frequency for this relatively fast channel. Some events (closure events) are in the same frequency range (lines 124-5). State the digital sampling rate.

Author Response

Dear Reviewer,

Thank you for your kind review our manuscript. We are very pleased that you liked it and that you appreciated its scientific value. We also grateful for pointing out the weaknesses of the manuscript, minor errors and ambiguities. Below we present answers to your suggestions or questions. We hope that we have made our statements exhaustively and that they will dispel your doubts.

Review 2

The manuscript reports a thorough and careful biophysical and biochemical investigation of the mitochondrial K(Ca)1.1/BK(Ca) channel found to be present in the inner membrane of a bronchial epithelial cell line. MitoBK(Ca) has been described in mitochondria from various other types of cells. This one presents some interesting elements of novelty concerning its kinetic behaviour in patch-clamp experiments and the molecular composition of its complex. I do have some questions and suggestions for improvement.

Is it possible to infer something about the high-activity variant of the channel from the effects of NS11021 on the control-activity variant? In other words, is it possible to attempt a comparison between the NS11021-treated control-activity channel and the untreated high-activity channel? Leads for discussion/speculation on this may come from the recent paper by Michael E. Rockman et al., 2020 (JGP) and the commentary by Jianmin Cui. Also, any speculation as to why NS1619 does not work?

Thank you for the comment. We added adequate text to the Discussion section.

Their findings induce the Authors to suggest that the control-activity channel may represent a complex containing the β3 subunit. This is likely to be correct, but then one wonders about the presence of non-negligible amounts of subunit β4 (Fig. 6). Is it possible that the high-activity channel represents a complex with β4? As the Authors write (lines 346-7) β4-comprising channels are resistant to IbTx. In Fig. 5 the manuscript shows inhibition of channel activity by IbTx, specifying that the results shown pertain to the control-activity channel. Have the Authors tested IbTx on the high-activity channel? If it turned out to be less sensitive to the toxin, this might support a tentative molecular identification and distinction between the two types.

Thank you for paying attention to not mentioning the sensitivity of the high activity channel of iberiotoxin. The group of high activity channel appears much less frequently than the control activity channel, which is why we performed some experiments only on control activity.

The sentence on lines 126-127 (“This finding may be due to an increased probability of channel opening, which causes the channel to remain closed for a very short time”) is obvious (of course the probability of opening is high if the mean closed time is short) and might as well be omitted.

Thank you for paying attention to the repetition of the information. Suitable text was removed.

Lines 193 and 323-6. Is not a slow and local diffusion of the toxin across the patch a more economic hypothesis to explain the partial inhibition of the channel?

Thank you for this comment. Indeed diffusion of the peptide might be potential explanation for observed effects. We added short notice in the text (see… section). At the moment we do not have clear data therefore we think that both options are possible.

Line 367: “while maintaining ΔΨ”: ?. NS11021 depolarizes mitochondria, right?

Yes, our results indicate the NS11021 induced depolarization of ΔΨ in human bronchial epithelial cells.

Lines 374-375: “is the attenuation of mitochondrial regulation in cells under oxidative stress”. Unclear. Explain

Thank you for paying attention to the incomprehensible fragment of the text. We would like to point out that the activation of mitochondrial potassium channels, including mitoBKCa channels, contributes to the reduction of the amount of reactive oxygen species during oxidative stress in the cell. We changed appropriate fragment.

Line 422: Axopatch not Axppatch

Thank you for noticing the spelling mistake in Axopatch.

Line 423. 1 kHz seems a rather low filter frequency for this relatively fast channel. Some events (closure events) are in the same frequency range (lines 124-5). State the digital sampling rate.

Thank you, this is important issue. Digital sampling rate was equal to 100 kHz. Appropriate text was added to the Materials and Methods section.